# On a Non-Discrete Concept of Prokaryotic Species

**DOI:** 10.3390/microorganisms8111723

**Published:** 2020-11-04

**Authors:** Juan M. Gonzalez, Elena Puerta-Fernández, Margarida M. Santana, Bhagwan Rekadwad

**Affiliations:** 1Instituto de Recursos Naturales y Agrobiología, Consejo Superior de Investigaciones Científicas, IRNAS-CSIC, Avda. Reina Mercedes 10, 41012 Sevilla, Spain; elena.puerta@irnas.csic.es; 2Centre for Ecology, Evolution and Environmental Changes (cE3c), Faculdade de Ciências da Universidade de Lisboa, Edifício C2, Campo Grande, 1749-016 Lisboa, Portugal; mmcsantana@fc.ul.pt; 3National Centre for Microbial Resource, National Centre for Cell Science, NCCS Complex, Savitribai Phule Pune University Campus, Ganeshkhind Road, Maharashtra State, Pune 411007, India; rekadwad@gmail.com

**Keywords:** species, discrete model, continuous model, taxonomy, taxon, microbial diversity, prokaryotes

## Abstract

The taxonomic concept of species has received continuous attention. A microbial species as a discrete box contains a limited number of highly similar microorganisms assigned to that taxon, following a polyphasic approach. In the 21st Century, with the advancements of sequencing technologies and genomics, the existence of a huge prokaryotic diversity has become well known. At present, the prokaryotic species might no longer have to be understood as discrete values (such as 1 or 2, by homology to Natural numbers); rather, it is expected that some microorganisms could be potentially distributed (according to their genome features and phenotypes) in between others (such as decimal numbers between 1 and 2; real numbers). We propose a continuous species concept for microorganisms, which adapts to the current knowledge on the huge diversity, variability and heterogeneity existing among bacteria and archaea. Likely, this concept could be extended to eukaryotic microorganisms. The continuous species concept considers a species to be delimited by the distance between a range of variable features following a Gaussian-type distribution around a reference organism (i.e., its type strain). Some potential pros and cons of a continuous concept are commented on, offering novel perspectives on our understanding of the highly diversified prokaryotic world, thus promoting discussion and further investigation in the field.

## 1. Introduction

Naming and classifying all living creatures is a requirement for science and humanity. Aristotle approached this around 2400 years ago [1]. Since then, different species definitions have been proposed, and all of them have received criticisms and opened debate among taxonomists from different disciplines [2,3]. The description of a concept of species includes similar organisms that are able to sexually reproduce among themselves [4]. This definition has been quite useful for macro-organisms, first looking into morphological features and, more recently, incorporating genomic characteristics, which is providing substantial assistance in the description of numerous case species [3] and the discovery of previously unknown intra-specific diversity [5,6].

In the microbial world, the current concept of species fails to adapt to the requirements for the appropriate and unambiguous description of similar groups and clades of microorganisms [1,7,8,9,10]. Microorganisms generally reproduce asexually, generating clonal cells. Thus, the proposal of co-reproduction is inadequate to classify prokaryotic cells. Besides, the morphology, phenotype and functional features are insufficient to unambiguously differentiate among prokaryotes [7,11]. Ecological distinguishing features, irreversible divergence among different taxa and consistent cohesiveness within a cluster are unique characteristics of species [12]. Species have multifarious features irreversibly raised by metapopulation lineages [13]. The use of genomic approaches is required to complement phenotypic characteristics, defining what has been established as a polyphasic approach mandatory for the description of novel prokaryotic species [8,10]. On the other hand, prokaryotes are incredibly diverse, presenting a multitude of different types [10,14] and showing an experimentally unmeasurable level of diversity [15,16,17]. The taxonomic analyses and phylogeny of prokaryotic diversity have been based on 16S rRNA gene sequencing technologies, which have boosted our current understanding of the diversity in the microbial world and keep providing us with highly valuable information on prokaryote distribution and ecology [15,18,19]. The rapid development of next-generation sequencing platforms has bloomed in the field of genomics (single taxon) and metagenomics (complex microbial communities), providing an enormous amount of information leading to a new genome-based perspective on the huge diversity of the prokaryotic world. For instance, there is on-going research on intra-species diversity and its limits [10,17,20,21], mostly based on the analysis of genome sequences. Public culture repositories hold growing numbers of strains (i.e., isolated, identified and further studied), and many of them have their genomes fully sequenced. An example, for instance, is the species *Escherichia coli*, with nearly 20,000 strains with their genome sequenced (NCBI Microbial Genome database), which includes a previously reported variability reaching up to 50% of their genomes [22]. Some other bacterial genera represent highly similar prokaryotic cells (i.e., *Escherichia*, *Shigella*, *Salmonella*), which remain difficult to differentiate. For other large prokaryotic genera, such as *Streptomyces*, *Clostridium*, *Bacillus*, etc., the differentiation among their species and strains often represents a compromising challenge [23,24,25].

The incorporation of genomics as novel variables in taxonomy, even overcoming the limitation of prokaryotic culturability for species delineation [7,15], will streamline species classification [10,26]. Nevertheless, in this field, the existing heterogeneity among strains and clonal lines has been demonstrated to be far from understood, and often leads to continuously growing pan-genomes [17,22,27,28]. Besides, the regulatory mechanisms of gene expression can lead to heterogeneity and epigenetic differentiation, as already demonstrated within clonal cultures [29], leading to further complexity when unifying phenotypic and genotypic information.

On top of that complexity generated by the huge diversity existing among the prokaryotes, the current taxonomical methodologies have incorporated scarce statistical approaches in defining the limits and the variability of a prokaryotic species. This should be considered, and it is essential to objectively classify a newly isolated strain, clone or cultivated prokaryotic cell line within a species, or as a newly proposed species or taxon. Prokaryotic taxonomists accept a 70% DNA–DNA hybridization threshold for species classification [30]. Other relatedness thresholds widely used are, for instance, 5 °C or less ΔT_m_ [30], and Multi-Locus Sequence Typing [11]. This has been a general rule of thumb applied for the proposal of novel species or the classification of a novel strain within a previously defined prokaryotic species. Similarly, more recently ANI (average nucleotide identity) analysis [1,31] of complete or draft genome sequences has been marked at 95% similarity to determine the strict classification of a prokaryote within a species. Neither of these thresholds consider statistical limits for species limits and variability. Numerous cases of strains or species do not fit within these thresholds, or overlap their characteristics, similarities or features. The known examples are, among many others, the species of the genera *Escherichia* and *Shigella*, which present high similarity among them. Similar problematics have been mentioned in some eukaryotes (e.g., fungi) when morphological features and genetic markers are limited to species delineation [32].

From the above, one could start envisioning the complexity of a proper system for prokaryote description and classification, which is under debate in taxonomic forums [8,10]. Thus, novel proposals to overcome these handicaps are required. The aim of this study is to propose a novel framework, and some of its (dis)advantages and consequences, representing a statistically solid perspective to understand the range limits of the prokaryotic species and to adapt to the current knowledge on prokaryote diversity and its increasingly known heterogeneity.

## 2. The Continuous Concept of Species

The existence of a huge prokaryotic diversity suggests that different microorganisms might present similar phenotypic features, and some of them can even show high genomic similarity. Previous publications have suggested the potential for a continuous prokaryotic species concept. Some examples, without the aim of presenting a full listing of contributions, can be mentioned. Gevers et al. [9] suggested that among some of the issues to be resolved, one of them is the difficulty of delineating groups and clusters within a continuous spectrum of genotypic variation. The existence of discrete prokaryotic species has been questioned [1,10], and the observation of a genetic continuum (lack of clusters) has been pointed out for several bacterial groups (i.e., Burkholderiaceae and Shewanellaceae) [9,10]. Another example is the case of some ubiquitous Planctomycetes whose phenotypic and genomic features present a challenge for their classification into species [33,34,35]. This information has led to the suggestion that “bacteria may frequently show a continuum of genetic diversity in nature” [10]. Additional research is required, but the increasing diversity and heterogeneity being observed within the prokaryotes progressively supports the concept for a continuity among prokaryotic species, even if clusters of similarity are delineated.

The vision of a prokaryotic world formed by discrete species might be a result of the bias introduced as a result of limited sampling of the actual diversity existing in nature. Atypical or rare isolates are often discarded, which negatively influences our full understanding of the continuity of diversity among prokaryotes [10]. Whereas in nature, the huge diversity and heterogeneity has been demonstrated to indicate continuity rather than completely independent and discrete units, additional sampling, isolation and genomic analyses will complete our spectrum of knowledge and contribute significantly to a better understand the prokaryotic world. Specifically, the continuous concept offers the potential for novel classification strategies, above all aimed at solving the current issue in the difficulty of unambiguously classifying all the prokaryotes thriving in nature.

The discrete and continuous concepts of prokaryotic species are not contradictory. The continuous concept is just a product of a more detailed knowledge acquired in the last years. This is leading to re-definitions as a result of the observed huge diversity of prokaryotes, the impressive advancement of genome sequencing platforms and the exponentially increasing genome information available. We understand that the concept of prokaryotic species should adapt to the current knowledge in the field, incorporating novel procedures for the unambiguous classification of bacteria and archaea, and with provisions for the future to be able to incorporate the increasing diversity (cultured and uncultured) that will soon be discovered and the enormous amounts of phenotypic and genomic information that we are becoming able to access.

One should understand the continuous concept of prokaryotic species as a cluster of phenotypic and/or genomic features, or information that groups together a number of strains, varieties or isolates due to their similarity. This is not much different from the classic, discrete concept, but incorporates a range of limits for the species, as well as a variability within, and a high diversity in and out of, those limits. The provision of statistical limits for a species and the acceptance of the potential existence of similar cells or strains out of the statistically delimited boundaries are key concepts that, up to today, have been mostly left aside in prokaryotic taxonomy, in spite of their known existence, the increasingly understood large diversity among the prokaryotes [10,18], and their already recognized incredible heterogeneity (genomic and epigenetic) [29,36,37]. Figure 1 visualizes these differences in the species concept as discrete units (Figure 1A) or continuous diversity (Figure 1B). The continuous concept of prokaryotic species should establish statistical-based thresholds to define a species, accounting for its variability and heterogeneity within a highly diverse and complex scenario of multiple, either known or yet to be discovered, strains, variants and prokaryotic cells. Thus, along a continuum of features (considering a polyphasic characterization), represented by known or yet unknown prokaryotes, a species should be considered as a cluster formed by a set of strains, variants or cells showing high similarity within a statistically meaningful range. Those cells presenting dissimilarities outside those limits would fill up the feature continuum and, as more knowledge is gathered, they can indicate future species to be delineated, but which are currently restricted by bias or scarce sampling, and are in need of further research.

## 3. Method

The method used to analyze the example species shown in this study is as follows. The genome sequences available for the analyzed species were retrieved from the National Center of Biotechnology Information (NCBI) Microbial Genomes database (https://www.ncbi.nlm.nih.gov/genome/browse#!/prokaryotes/). For each species, its type strain was selected as the reference organism except if noted otherwise. Within a species, the genomes were pairwise compared to the reference strain and the genomic distance (% dissimilarity) was calculated based on the average nucleotide identity (ANI) as described by Richter and Rosselló-Móra [1]. Briefly, query genomes were divided into 500 nucleotide fragments, which were used to search against the whole genome sequence of the reference genome using the blastn algorithm [31]. ANI was estimated, as the average of the genome fragments, by the percentage of identity (resulting from blastn) times the aligned sequence length divided by fragment length for the genome fragments. Genomic distance was expressed as the percentage of dissimilarity (100-ANI) between query and reference genome, with the type strain showing zero dissimilarity (i.e., it was identical) to itself. For each species, the frequency distribution of those dissimilarities was calculated using a standard office-ware spreadsheet. The Appendix A shows the list of the example species analyzed in this study, the type strains, and the number of genomes available at NCBI for each species.

## 4. Potential Issues, Advances and Solutions

Attempts to group or cluster entities within a continuum of characteristics have been a major drawback in accepting the existence of continuum diversity among the prokaryotes. Nevertheless, as the known prokaryotic diversity is deciphered and our understanding on the subject increases, we reach the point where at discrete, independent microbial units (i.e., species) are difficult to sustain. The continuity concept does not interfere with setting limits or boundaries for the classification of the members of a species. Mathematicians constantly define sets and subsets within the real numbers. It is expected that a species represents a cluster of cells, strains or variants (representing the intraspecific variability) within some established limits (range of confidence) than can be classified to belong to that species because all the other cells (those off boundaries) are placed (following the same analysis) in the continuum spectrum beyond those limits. There is no reason to limit our understanding to discrete units, such as natural numbers (e.g., 1, 2, 3, and so on) in mathematics. Considering a continuum, one can better understand the actual reality in the microbial world (i.e., due to its huge diversity), such as the case of real numbers in mathematics, which reveals the existence of a continuum of endless possibilities (i.e., decimal numbers) in between two counting numbers. Microbiologists should face the challenge and adapt to the current knowledge and scenarios among prokaryotes.

Difficulty does not imply prohibition. Advances in the delineation of prokaryotic taxa must pick concepts from other sciences following the true purpose of multidisciplinary science. Mathematicians have been developing the study of distributions for centuries, and today there is a well-established theory of distributions (especially well studied for, as an example, Gaussian distribution), variability and limits of significance [38] that can be applied to the delineation of species among prokaryotes (Figure 2A). Microbiologists can take advantage of that solid statistical knowledge to provide well-defined, statistically solid approaches to species delineation within an accepted continuum, or near-continuum, of diversity within the prokaryotes.

Figure 2 shows the frequency distribution plot for additional prokaryotic species to confirm the universality of the proposed continuous concept of species. As expected, the curves are bell-shaped. These curves’ peaks correspond to the cluster that should define a species. By applying the mathematics around these types of bell-shaped or Gaussian distributions, one should be able to clearly define the variability existing within a bacterial species. Thus, a properly delineated species should include the statistics of its mean and variance (deviation from the mean), which adequately define the distribution curve and easily set an interval of confidence (or limits of acceptance) for the species. Different species will present separate clusters, which should be differentiated from each other. For instance, Figure 1B shows the comparison of the closely related species *Shigella flexneri* and *Escherichia coli*. Figure 3 compares a case of a single-peak, bell-shaped curve (corresponding to *Shigella sonnei*) and a multi-modal curve (corresponding to *Shigella boydii*), showing three clusters within the species. Some other examples of multi-modal curves can be observed in Figure 2. The multi-modal cases would suggest that the known spectrum for a species is represented by relatively distant clusters, and those strains positioned under each of the sub-peaks could be a potential target for their re-classification as different species. In Figure 3A, *S. sonnei* represents a uni-modal curve (some additional examples in Figure 2), suggesting a compact cluster in a well-defined bacterial species.

The use of statistical methods for the taxonomic delineation of species requires further attention. Today’s classic concept of species is likely a result of limitations in the understanding of the actual microbial diversity, as a consequence of scarce sampling. Adequate and more abundant sampling should provide a high number of strains for a good description of the statistical parameters that will delineate the species. In order to warrant a good species delineation, one should gather results from extensive and intensive sampling (i.e., a high number of strains for a species, if available, from different geographical areas) so that the statistics are able to provide significant data, thus minimizing biases due to limited sampling. The current limitations on strain availability or number of isolates negatively affect the understanding of microbiology and the description of species. 

Today, there is a strong initiative to fully incorporate genomics into species delineation [39]. It is recognized that all the phenotypic capabilities available to a cell should be reflected in its genome. Thus, genomes should be a major aspect in species definition [10]. Genome similarity analyses could be a major starting point for the classification and species delineation of prokaryotes within the proposed genetic continuum existing in nature. There are different approaches to determine the distance (similarity or dissimilarity) between bacteria, most of them based on their complete or partial genome sequence. The examples are ANI, Bray–Curtis dissimilarity, hierarchical clustering, non-metric multidimensional scaling (NMDS), principal components analysis (PCA) or principal coordinates analysis (PCoA), and other clustering methods [1,40,41,42] that could be applied to genotypic and phenotypic features. Novel approaches are being presented [43], and others might be under development.

Nevertheless, independently of the analyzed features, similar trends should be applied to construct a set of bell-shaped curves that will define a cluster corresponding to a species. Ideally, the polyphasic approach [7,8] should be applied and novel parameters and procedures could be developed to incorporate both phenotypic and genotypic features into a multiparametric approach so as to determine the similarity (or dissimilarity) of strains within a species, and to establish the range setting the limits that define each species. In this respect, different omics (genomics, proteomics, metabolomics, etc.) and as many cellular characteristics as possible (e.g., morphology, cell wall and fatty acid composition, enzyme and metabolic activities, etc.) should be ideally integrated into the delineation of a species and its definition within the proposed microbial continuum.

## 5. Consequences and Research Lines

One of the major goals of taxonomy should be to determine the variation allowed within a species or taxa. This should be performed based on a statistical procedure that considers descriptive statistics, such as average and variance (deviation from the mean), mode, and median, which would assist in defining the range of distances (or dissimilarities) from a reference strain (ideally, the type strain) within a prokaryotic species.

The different consequences of these analyses can be observed in Figure 2 and Figure 3. For instance, the mode for a bell-shaped species distribution should include the reference strain of choice, and this should be assimilated into the type strain for that species (Figure 3A). Thus, a type strain could be easily defined, and for many species this comparative procedure could be used to facilitate the proposal of the ideal representative type strain for each species. Another common scenario is that of a multi-modal distribution for a species (Figure 2B), which is a consequence of limited sampling or the need for a reclassification of strains in different taxa. 

A major advantage of large data sets representative of the actual diversity and heterogeneity within a species is that future estimates over time-series or spatial scales can provide evidence of the potential shifts or trends in a species’ features. The detection of novel diversity and putative speciation processes could lead to species differentiation and the extinction or preservation of the representatives of a species or taxa. Currently, long-term preservation in public culture collections has revealed strain biases [44,45]. This phenomenon remains poorly understood, but it might be a consequence of the continuous evolution and adaptive processes in the prokaryotes. The use of statistics in microbial taxonomy will certainly allow us to determine levels of significance, so as to distinguish differential clusters as different species and provide a whole new level of definition in microbial taxonomy. As of today, the processes of speciation, diversification, and the preservation or generation of novel species remain barely understood in the prokaryotic world. Although it is expected that prokaryotes follow similar rules to those described for eukaryotes, the high dynamism expected in the prokaryotes (including active horizontal gene transfer among different taxa) and the plasticity of their genomes, as previously suggested [46,47,48], underline the interest in these research topics to better understand the past, present and future of microbial phylogeny and evolution.

## 6. Perspectives and Future Trends

Increasing efforts in the detection, isolation and culturing of microorganisms, and the exponential advances in next-generation sequencing platforms, are allowing us to reach a much deeper knowledge of the actual prokaryotic diversity and its spectrum of variability. Besides the new and exciting potential research trends mentioned above, the future initiatives in this field should be focused on the development of novel and unified comparative approaches to determine phenotypic and genomic distances (leading to the description of polyphasic distance metrics). Although the use of genomic information is a must in species comparison, past taxonomic experience continues to highlight the importance of the phenotype in determining species differentiation. Future research should, as well, be able to discriminate the taxonomic interest and the limitations of the phenotype versus the importance of epigenetic heterogeneity in prokaryotes. These novel research trends need to be evaluated so as to select the relevant information related to phenotypes, genomes and their intrinsic intra-specific heterogeneity. Again, the role of statistical methods would significantly contribute to delineating species, including the range of variability introduced by the members of a species. Microbial taxonomy and the concept of species must incorporate the recently acquired information and knowledge from current research trends, and adapt to the new changes and challenges that are about to come in the near future, which will certainly restructure the way microbiology has been understood so far.

## Figures and Tables

**Figure 1 microorganisms-08-01723-f001:**
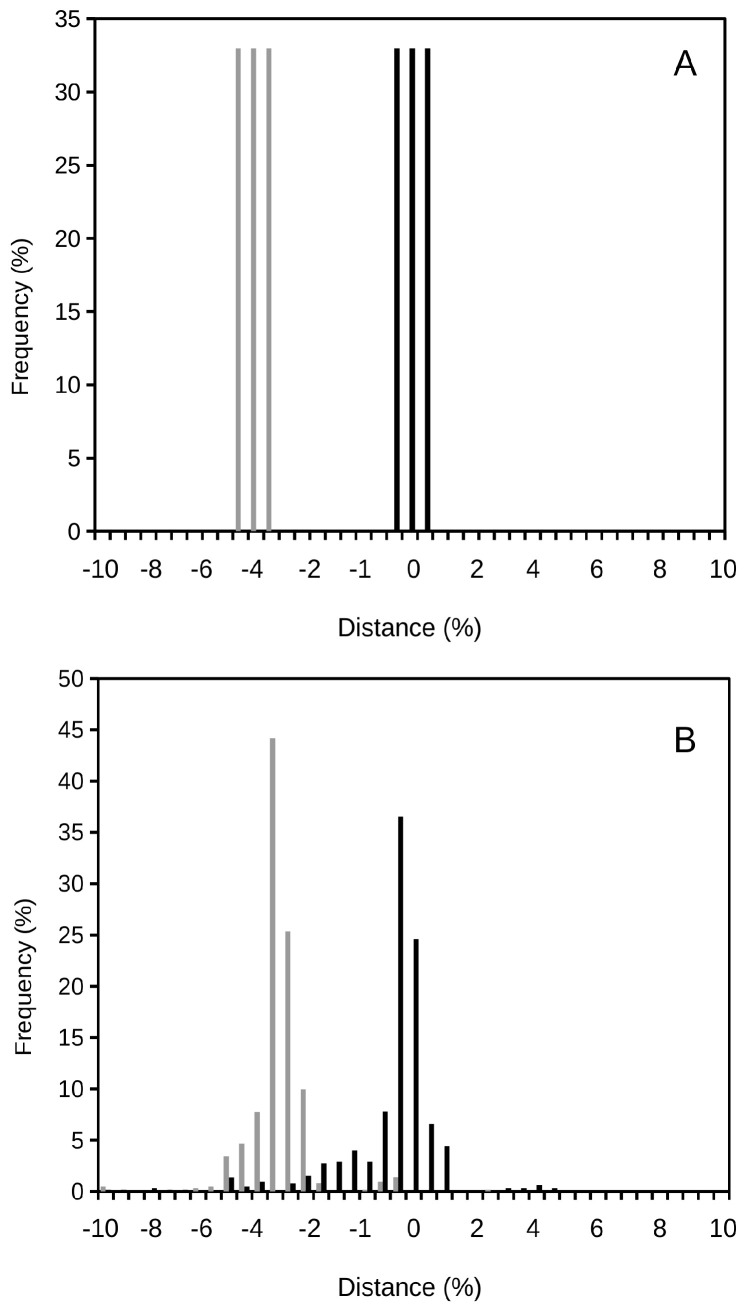
A visual comparison of the classical discrete species concept (**A**) and the suggested continuous species concept (**B**) for prokaryotes when plotting distance among strains versus their frequency distribution. In the classical, discrete concept, no strains or cells are considered at the distances existing in between species. Note that the continuous concept considers the existence of (either known or to be discovered) diversity and heterogeneity among the prokaryotes. Furthermore, one should consider the bell-shaped distribution, which will allow accurate statistical analyses to be applied to the taxonomic evaluation of prokaryotic species. The continuous concept also depends on the clustering of strains or variants around a reference strain, but simultaneously recognizes a continuum of features which includes the existence, or potential existence, of other prokaryotes off the species boundaries. In (**B**), the examples of *Shigella flexneri* (black) and its comparison with *Escherichia coli* (grey) are shown. Distance represents the percentage of dissimilarity against the reference genome.

**Figure 2 microorganisms-08-01723-f002:**
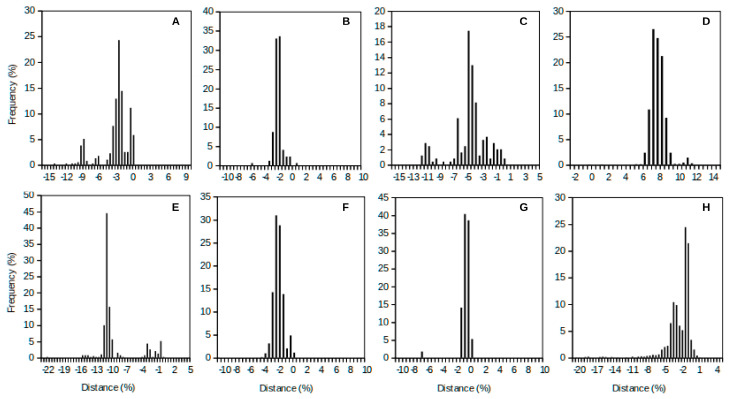
Eight additional examples showing the plots of the frequency distribution versus distance to the type strain of each species. The analysis was carried out for the species: *Bacillus subtilis* (**A**), *Bacteroides fragilis* (**B**), *Clostridium botulinum* (**C**), *Helicobacter pylori* (**D**), *Rhizobium leguminosarum* (**E**), *Streptococcus pyogenes* (**F**), *Sulfolobus acidocaldarius* (**G**), and *Vibrio cholerae* (**H**). The displacement of the main peak to one of the sides, away from zero distance, might suggest that a better reference (i.e., type strain) should be established. Distance represents the percentage of dissimilarity against the reference genome.

**Figure 3 microorganisms-08-01723-f003:**
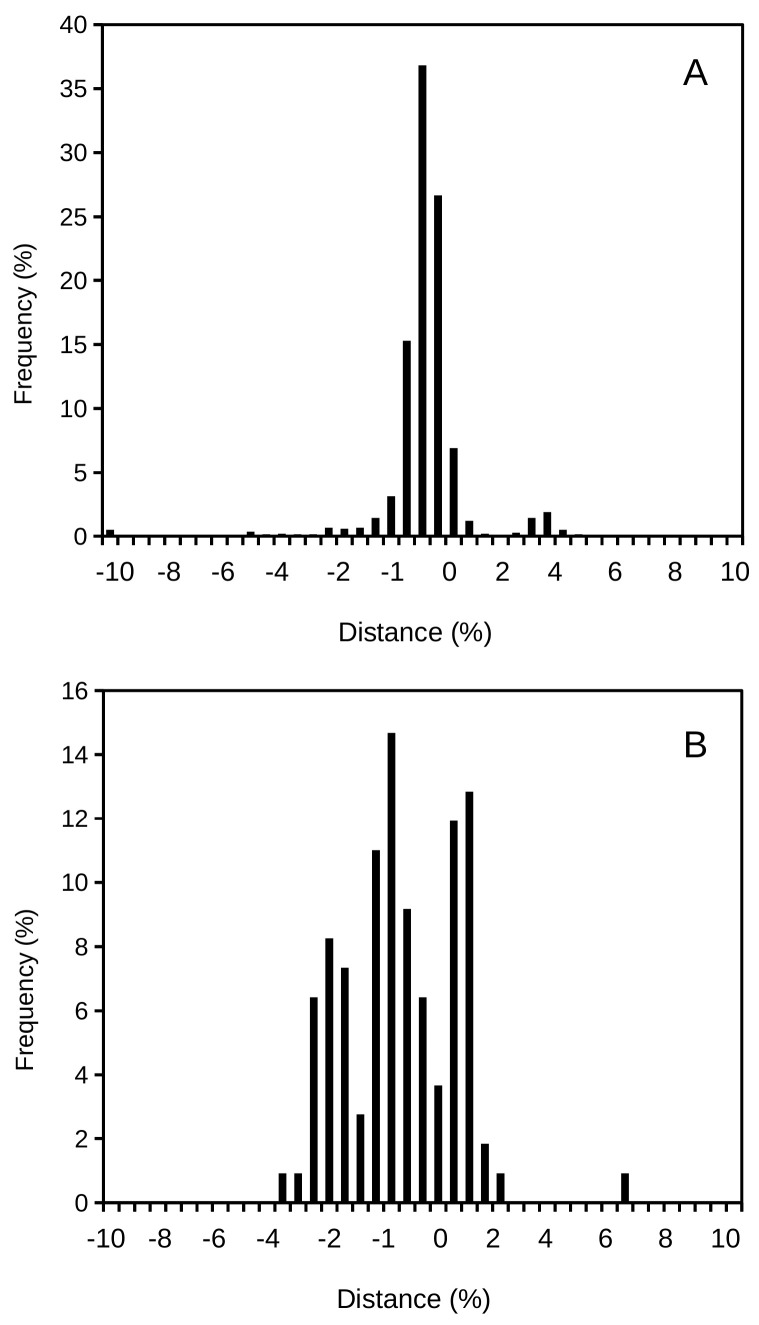
Examples of uni-modal and multi-modal distributions shown with the examples of *Shigella sonnei* (**A**) and *Shigella boydii* (**B**), respectively, when plotting distance among strains versus their frequency distribution. Note the multiple peaks observed within the spectrum of the species *S. boydii* while *S. sonnei* shows a single peak. Distance represents the percentage of dissimilarity against the reference genome.

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
