# Peer review of "On a Non-Discrete Concept of Prokaryotic Species"

_microorganisms, 2020, doi:10.3390/microorganisms8111723_

Round 1

Reviewer 1 Report

The manuscript, by Juan M. Gonzalez, Elena Puerta-Fernández, Margarida M. Santana, Bhagwan Rekadwad, sent to Microorganisms – 949390, titled „On a non-discrete concept of prokaryotic species” is of perspective type.

The authors proposed a concept of highly divered prokariotic species for microorganisms, named continous. This conception considers a species to be delimited by the distance of a range of variable features following a Gaussian-type distribution around a reference as strain, organism etc. The work fits well as to the current literature as to ongoing debate concernig the search of the proper system for the prokaryote description and classification.

The presented continous concept is mostly clearly defined and the methods employed to test it are rather convincing. The article is well written I recommend this article to be published in Microorganisms under one condition. The results presented in Figure 1 based on the study on Shigella flexneri and Escherichia coli are not enough to be fully convinced. For the reviewer it should be added some more analysis based on the other organisms/strains reflecting the similar tendency. Therefore Figure 1 should be rebuilt.

The lines of the manuscript are not numbered.

The type of the paper is not specified.

Author Response

Reviewer #1.

Response. Thanks to this Reviewer for its positive recommendation. This Reviewer presents one requirement which is the inclusion in the ms of analyses for some additional species to show the generality of the arguments.

We have added the analysis (i.e., frequency distribution vs. distance curves) for 8 additional species (new Figure 2). We have also added Table S1 (line 165) as supplementary material where we list the species used in the examples and the number of genomes available at NCBI (Microbial Genomes database; https://www.ncbi.nlm.nih.gov/genome/browse#!/prokaryotes/) for each of them. Besides the previously included species, the additional species are included in a new Figure (Figure 2). They are: Bacillus subtilis, Bacteroides fragilis, Clostridium botulinum, Helicobacter pylori, Rhizobium leguminosarum, Streptococcus pyogenes, Sulfolobus acidocaldarius, and Vibrio cholerae.

We believe now the universality of the proposal is confirmed and we hope we have satisfied the Reviewer’s requirement.

Response. Sorry, line numbers have now been added to the ms.

Response. Sorry. The ms did not include the type of manuscript in the front page. It was indicated in the online submission. We have now added this in the front page following the Microorganisms template.

Reviewer 2 Report

This manuscript described on a non-discrete concept of prokaryotic species.

Whether or not it applies to all cases in prokaryotic taxonomy is controversial, but it can be believed that there are certainly cases where it is necessary to consider the non-discrete concept. However, the authors should be presented concrete examples on the basis of the data that the reader obtainable such as gene database. If the authors possess your own data, they should be disclosed for reproduction by a third party. In addition, the analyzed procedure for analyses of bacteria and statistics used for the Figs should be presented.

Since many descriptions are conceptual and considered to be very difficult for the reader to understand, explanations based on concrete examples should be required. Bacterial taxonomy is based on a certain amount of history and experience, and it is thought that there is a problem in raising a problem only with conceptual content.

To show the proof of this proposal accurately and concretely, the whole genome analyzed ca. 100 strains collected from different sources that recognized as the same species by DDH and ANI. To observed genetic variability, the collected stains should be analyzed by another reliable method (e.g. TOF mass analysis). However, there is no description on the authors' views on this point.

Although it is not possible to draw a conclusion without collecting all the strains that can be isolated, it is thought that there are species that are far from others that do not require this non-discrete concept, but this is also not specifically described.

Some of the presented Figures are based on the authentic real data. Therefore, the authors should described the sources of the data used for the calculations. The methodology of the analysis should also be described. There is no description of the unit for distance horizontal axis in the Figs.

Author Response

Reviewer # 2.

Response: We have included in the ms a section describing the methods (lines 154-166) that we have used for the analysis of several species as examples of the continuous species perspective. The methods are simple but they were reduced too much in the previous version of the ms. Now, in this new version, we have incorporated a more detailed description which also respond some of the comments explained below.

Response: We had attempted to highlight repeatedly a need for a polyphasic approach and new methods for the differentiation of clusters defining the species. We agree with this Reviewer and taxonomy is and will continue needing history and experience. As for Reviewer # 1, in order to show the generality of the proposed concept of species, we have incorporated the analysis of several new species, which are listed in Table S1 (Supplementary Material) and the results presented in the new Figure 2.

Response: Thank you for the comment because it is always good to remark the need for using multiple methodologies to differentiate species. We do not attempt to change the classification of the species, we propose a new concept and framework that could better incorporate the novel knowledge on the existing huge prokaryotic diversity and adapt better to the new requirements of the prokaryotic concept of species. The new perspective will aim to better understand the concept of species following current knowledge and requirements. Because we do not attempt to redefine each of the species, there is no need to perform additional classification procedures. We also mentioned the need to incorporate different approaches into a true polyphasic approach to define the continuous nature of the concept of prokaryotic species. This approach should include phenotypic and genotypic features in agreement to current requirements for taxonomic classification of species. We do not perform a reclassification of the species, we propose a new concept to delineate the existing (and the new ones to be discovered) prokaryotes under a perspective that considers the broad diversity and heterogeneity within the prokaryotic world.

Response: The concept of species must be unique because it must be the framework to include the known, and those to be discovered, microorganisms so that we can distinguish species and cluster the different strains in or off the limits of each species. A species can include different strains or variants. Even if we only know a single strain for a species, it does not imply that there are not more strains or cells that could belong to that species. This is a typical example of what has been considered limited sampling which means that only one strain has been detected but it is expected that many more will be out there, we just have not found them yet. Limiting sampling is the major problem that Microbiology is facing today and future and novel perspectives must incorporate the current perspective that microbial diversity represent (or tend to) a continuum diversity rather that to discrete, independent units. We have shown the homology of this concept with the difference between Natural numbers (1, 2, 3, and so on, a discrete concept) and Real (decimal) numbers (with infinite decimal numbers between 1 and 2, and between 2 and 3 and so on; a continuous concept).

Response: The new section on methods (lines 154-166) describes were the genomes can be collected from and how these have been processed. The procedure used in this study is simple and has been previously proposed (Richter and Rosselló-Móra, 2009). It is based on ANI estimations (lines 161). The genomes used in this study are available to the public at the NCBI, Microbial Genomes database (https://www.ncbi.nlm.nih.gov/genome/browse#!/prokaryotes/).

Round 2

Reviewer 2 Report

The manuscript described on the non-discrete concept of prokaryotic species has been revised.

The revised manuscript has become much more acceptable than the previous version.

Please describe definition of Figure’s horizontal axis (distance) and please explain how to calculate the distance value briefly.

P2 L19: “Escherichia coli” should be described by italics.

Author Response

The manuscript described on the non-discrete concept of prokaryotic species has been revised.

The revised manuscript has become much more acceptable than the previous version.

Please describe definition of Figure’s horizontal axis (distance) and please explain how to calculate the distance value briefly.

Response:

Thank you. We have introduced a brief description of the protocol to calculate genomic distances (lines 162-167 in version with track changes and lines 160-166 in the final version). On the figures, we now mention that distance in the X-axis of the figures represents the percentage of genomic dissimilarity (Figure legends; lines 166-167 in version with track changes and lines 165-166 in the final version).

P2 L19: “Escherichia coli” should be described by italics.

Response:

We have written “Escherichis coli” in italics (line 62).